# Learning to solve the credit assignment problem

**Benjamin J. Lansdell**
Department of Bioengineering
University of Pennsylvania
Philadelphia, PA 19104
lansdell@seas.upenn.edu

**Prashanth R. Prakash**
Department of Bioengineering
University of Pennsylvania
Philadelphia, PA 19104

**Konrad P. Kording**
Department of Bioengineering
University of Pennsylvania
Philadelphia, PA 19104

## Abstract

Backpropagation is driving today's artificial neural networks. However, despite extensive research, it remains unclear if the brain implements this algorithm. Among neuroscientists, reinforcement learning (RL) algorithms are often seen as a realistic alternative. However, the convergence rate of such learning scales poorly with the number of involved neurons. Here we propose a hybrid learning approach, in which each neuron uses an RL-type strategy to learn how to approximate the gradients that backpropagation would provide. We show that our approach learns to approximate the gradient, and can match the performance of gradient-based learning on fully connected and convolutional networks. Learning feedback weights provides a biologically plausible mechanism of achieving good performance, without the need for precise, pre-specified learning rules.

## 1 Introduction

It is unknown how the brain solves the credit assignment problem when learning: how does each neuron know its role in a positive (or negative) outcome, and thus know how to change its activity to perform better next time? Biologically plausible solutions to credit assignment include those based on reinforcement learning (RL) algorithms [4]. In these approaches a globally distributed reward signal provides feedback to all neurons in a network. However these methods have not been demonstrated to operate at scale. For instance, variance in the REINFORCE estimator scales with the number of units in the network. This drives the hypothesis that learning in the brain must rely on additional structures beyond a global reward signal.

In artificial neural networks, credit assignment is performed with gradient-based methods computed through backpropagation. This is significantly more efficient than RL-based algorithms. However there are well known problems with implementing backpropagation in biologically realistic neural networks. For instance backpropagation requires a feedback structure with the same weights as the feedforward network to communicate gradients (so-called weight transport). Yet such structures are not observed in neural circuits. Despite this, backpropagation is the only method known to solve learning problems at scale. Thus modifications or approximations to backpropagation that are more plausible have been the focus of significant recent attention [8, 3]. Notably, it turns out that weight transport can be avoided by using fixed, random feedback weights, through a phenomenon called feedback alignment [8]. However feedback alignment does not work in larger, more complicated network architectures (such as convolutional networks).

33rd Conference on Neural Information Processing Systems (NeurIPS 2019), Vancouver, Canada. Real Neurons & Hidden Units Workshop.

Here we propose to use an RL algorithm to train a feedback system to enable learning. We propose to use a REINFORCE-style perturbation approach to train a feedback signal to approximate what would have been provided by backpropagation. We demonstrate that our model learns as well as regular backpropagation in small models, overcomes the limitations of fixed random feedback weights ("feedback alignment") on more complicated feedforward networks, and can be utilized in convolutional networks. Our method illustrates a biologically realistic way the brain could perform gradient descent-like learning.

## 2 Method

Let an $N$ hidden-layer network be given by $\hat{\mathbf{y}} = f(\mathbf{x}) \in \mathbb{R}^p$, composed of a set of layer-wise summation and non-linear activations $\mathbf{h}^i = f^i(\mathbf{h}^{i-1}) = \sigma\left(W^i \mathbf{h}^{i-1}\right)$, for hidden layer states $\mathbf{h}^i \in \mathbb{R}^{n_i}$, non-linearity $\sigma$ and with input $\mathbf{h}^0 = \mathbf{x}$ and output $\mathbf{h}^{N+1} = \hat{\mathbf{y}}$. Define $\mathcal{L}$ as the loss function $\mathcal{L}(\mathbf{x}, \mathbf{y})$, where the data $(\mathbf{x}, \mathbf{y}) \in \mathcal{D}$ are drawn from a distribution $\rho$. Our aim is then to minimize: $\mathbb{E}_\rho\left[\mathcal{L}(\mathbf{x}, \mathbf{y})\right]$. Backpropagation computes the error signal $\tilde{\mathbf{e}}^i$ in a top-down fashion:

$$\mathbf{e}^i = \begin{cases} \partial\mathcal{L}/\partial\hat{\mathbf{y}} \circ \sigma'(W^i \mathbf{h}^{i-1}), & i = N+1; \\ \left((W^{i+1})^T \mathbf{e}^{i+1}\right) \circ \sigma'(W^i \mathbf{h}^{i-1}), & 1 \leq i \leq N \end{cases}. \tag{1}$$

Let the loss gradient term be denoted as $\lambda^i = \frac{\partial\mathcal{L}}{\partial\mathbf{h}^i} = (W^{i+1})^T \mathbf{e}^{i+1}$. Here we replace $\lambda^i$ with an approximation, with its own parameters to be learned: $\lambda^i \approx \mathbf{g}(\mathbf{h}^i, \tilde{\mathbf{e}}^{i+1}; B)$, for parameters $B$. We will use $\tilde{\mathbf{e}}^i$ to denote the gradient signal backpropagated through the synthetic gradients, and $\mathbf{e}^i$ for the true gradients. To estimate $B$ we use stochasticity inherent to biological neural networks. For each input each unit produces a noisy response: $\mathbf{h}_t^i = \sigma\left(\sum_k W_{\cdot k}^i \mathbf{h}_t^{i-1}\right) + c_h \xi_t^i$, for independent Gaussian noise $\xi^i \sim \nu = \mathcal{N}(0, I)$ with standard deviation $c_h > 0$. This then generates a noisy loss $\tilde{\mathcal{L}}(\mathbf{x}, \mathbf{y}, \xi)$ and a baseline loss $\mathcal{L}(\mathbf{x}, \mathbf{y}) = \tilde{\mathcal{L}}(\mathbf{x}, \mathbf{y}, 0)$. We will use the noisy response to estimate gradients, that then allow us to optimize the baseline $\mathcal{L}$. This is achieved by linearizing the loss: $\tilde{\mathcal{L}} \approx \mathcal{L} + \frac{\partial\mathcal{L}}{\partial h_j^i} c_h \xi_j^i$. This gives $\mathbb{E}((\tilde{\mathcal{L}} - \mathcal{L})c_h \xi_j^i | \mathbf{x}, \mathbf{y}) \approx c_h^2 \frac{\partial\mathcal{L}}{\partial h_j^i}|_{\mathbf{x}, \mathbf{y}}$, with expectation taken over the noise distribution $\nu(\xi)$. Thus we obtain an estimator of the loss gradient: $\hat{\lambda}^i := (\tilde{\mathcal{L}}(\mathbf{x}, \mathbf{y}, \xi) - \mathcal{L}(\mathbf{x}, \mathbf{y}))\frac{\xi^i}{c_h}$.

## 3 Applications

### 3.1 Fully connected networks solving MNIST

To demonstrate the method can be used to solve simple supervised learning problems we use node perturbation with a four-layer network and MSE loss to solve MNIST (Fig. 1). We approximate loss gradients as follows: $\mathbf{g}(\mathbf{h}^i, \tilde{\mathbf{e}}^{i+1}; B) = (B^{i+1})^T \tilde{\mathbf{e}}^{i+1}$. The feedback parameters $B^{i+1}$ are estimated by solving the least squares problem: $\hat{B}^{i+1} = \operatorname{argmin}_B \mathbb{E}\|B^T \tilde{\mathbf{e}}^{i+1} - \hat{\lambda}^i\|_2^2$, where $\hat{\lambda}$ is the perturbation-based estimator derived above. $B$ is updated with each mini-batch using stochastic gradient-descent to minimize this loss.[1] Updates to $W^i$ are made using the synthetic gradients $\Delta W^i = \eta \tilde{\mathbf{e}}^i \mathbf{h}^{i-1}$, for learning rate $\eta$. We observed that the system is able to provide a close correspondence between the feedforward and feedback matrices in both layers of the network (Fig. 1a). The relative error between $B_i$ and $W_i$ is lower than what is observed for feedback alignment, suggesting that this co-adaptation of $W_i$ and $B_i$ is indeed beneficial. We observe that the alignment (the angle between the estimated gradient and the true gradient, proportional to $\mathbf{e}^T W B^T \tilde{\mathbf{e}}$) is lower for node perturbation than for feedback alignment (Fig. 1b). Recent studies have shown that sign congruence of the feedforward and feedback matrices is all that is required to achieve good performance [10]. Here the sign congruence is also higher in node perturbation (Fig. 1c). Finally, the learning performance of node perturbation is comparable to backpropagation (Fig. 1d) – achieving close to 3% test error. These suggest node perturbation for learning feedback weights can be used in deep networks.

---

[1]We can prove this converges to the true weights when applied to either non-linear shallow network or a deep linear network, though these results are omitted here for brevity. These details are provided in the supplementary material.

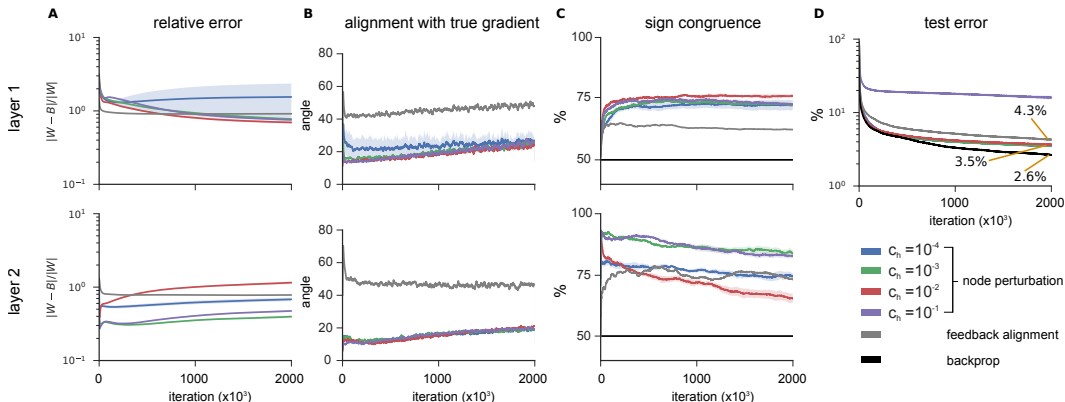

Figure 1: Node perturbation in small 4-layer network (784-50-20-10 neurons), for varying noise levels $c$, compared to feedback alignment and backpropagation. (A) Relative error between feedforward and feedback matrix. (B) Angle between true gradient and synthetic gradient estimate for each layer. (C) Percentage of signs in $W_i$ and $B_i$ that are in agreement. (D) Test error for node perturbation, backpropagation and feedback alignment. Curves show mean plus/minus standard error over 5 runs. Hyperparameters found through random search.

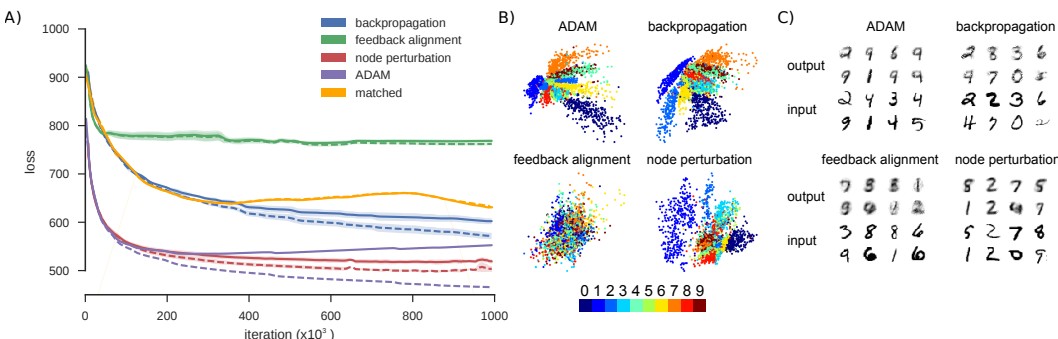

Figure 2: Results with five-layer MNIST autoencoder network. a) Mean loss plus/minus standard error over 10 runs. Dashed lines represent training loss, solid lines represent test loss. b) Latent space activations, colored by input label for each method. c) Sample outputs for each method. Hyperparameters found through random search.

## 3.2 Auto-encoding MNIST

A known shortcoming of feedback alignment is in auto-encoding networks with tight bottleneck layers [8]. To see if our method has the same shortcoming we examine a simple auto-encoding network with MNIST input data (size 784-200-2-200-784, MSE loss). We also compare the method to the 'matching' learning rule [9], in which updates to $B$ match updates to $W$. As expected, feedback alignment performs poorly. Node perturbation actually performs better than backpropagation, and comparable to ADAM (Fig. 2a). In fact ADAM begins to overfit, while node perturbation does not. The matched learning rule performs similarly to backpropagation. These results are surprising at first glance. Perhaps, similar to feedback alignment, learning feedback weights strikes the right balance between providing a useful signal to learn, and constraining the updates to be sufficiently aligned with $B$, acting as a type of regularization [8]. The noise added when estimating the feedback weights may also serve to regularize the latent representation, as, indeed, the latent space learnt by node perturbation shows a more evenly distributed separation of digits. While, in contrast, the representations learnt by backprop and ADAM show more structure, and feedback alignment does not learn a useful representation at all (Fig. 2b,c). These results show that node perturbation is able to successfully communicate error signals through thin layers of a network as needed.

### 3.3 Convolutional neural networks solving CIFAR10

Finally we test the method on a convolutional neural network (CNN) solving CIFAR10. The CNN has the architecture Conv(3x3, 1x1, 32), MaxPool(3x3, 2x2), Conv(5x5, 1x1, 128), MaxPool(3x3, 2x2), Conv(5x5, 1x1, 256), MaxPool(3x3, 2x2), FC 2048, FC 2048, Softmax(10), with hyperparameters found through random search. For this network we learn feedback weights direct from the output layer to each earlier layer: $\mathbf{g}(\mathbf{h}^i, \tilde{\mathbf{e}}^{i+1}; \theta_i) = (B^{i+1})^T \tilde{\mathbf{e}}^N$ (similar to 'direct feedback alignment'). Here this was solved by gradient-descent. We obtain a test accuracy of 75.2%. When compared with fixed feedback weights (test accuracy of 72.5%) and backpropagation (test accuracy of 77.2%), we see it is advantageous to learn feedback weights. This shows the method can be used in a CNN, and can solve challenging computer vision problems without weight transport.

## 4 Discussion

Here we implement a perturbation-based synthetic gradient method to train neural networks. We show that this hybrid approach can be used in both fully connected and convolutional networks. By removing both the symmetric feedforward, feedback weight requirement imposed by backpropagation this approach is a step towards more biologically-plausible deep learning. In contrast to many perturbation-based methods, this hybrid approach can solve large-scale problems. We thus believe this approach can provide powerful and biologically plausible learning algorithms.

While previous research has provided some insight and theory for how feedback alignment works [8, 3, 2] the effect remains somewhat mysterious, and not applicable in some network architectures. Recent studies have shown that some of these weaknesses can be addressed by instead imposing sign congruent feedforward and feedback matrices [10]. Yet what mechanism may produce congruence in biological networks is unknown. Here we show that the shortcomings of feedback alignment can be addressed in another way: the system can learn to adjust weights as needed to provide a useful error signal. Our work is closely related to Akrout et al 2019 [1], which also uses perturbations to learn feedback weights. However our approach does not divide learning into two phases, and training of the feedback weights does not occur in a layer-wise fashion.

Here we tested our method in an idealized setting, however it is consistent with neurobiology in two important ways. First, it involves the separate learning of feedforward and feedback weights. This is possible in cortical networks where complex feedback connections exist between layers, and where pyramidal cells have apical and basal compartments that allow for separate integration of feedback and feedforward signals [5]. Second, noisy perturbations are common in neural learning models. There are many mechanisms by which noise can be measured or approximated [4, 7], or neurons could use a learning rule that does not require knowing the noise [6]. While our model involves the subtraction of a baseline loss to reduce the variance of the estimator, this does not affect the expected value of the estimator; technically the baseline could be removed or approximated [7]. Thus we believe our approach could be implemented in neural circuits. There is a large space of plausible learning rules that can learn feedback signals in order to more efficiently learn. These promise to inform both models of learning in the brain and learning algorithms in artificial networks. Here we take an early step in this direction.

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

## Supplementary Material

## A  Proofs

We review the key components of the model. Data $(\mathbf{x}, \mathbf{y}) \in \mathcal{D}$ are drawn from a distribution $\rho$. The loss function is linearized:

$$\tilde{\mathcal{L}} \approx \mathcal{L} + \frac{\partial \mathcal{L}}{\partial h_j^i} c_h \xi_j^i, \tag{2}$$

such that

$$\mathbb{E}\left( (\tilde{\mathcal{L}} - \mathcal{L}) c_h \xi_j^i | \mathbf{x}, \mathbf{y} \right) \approx c_h^2 \left. \frac{\partial \mathcal{L}}{\partial h_j^i} \right|_{\mathbf{x}, \mathbf{y}},$$

with expectation taken over the noise distribution $\nu(\xi)$. This suggests a good estimator of the loss gradient is

$$\hat{\lambda}^i := (\tilde{\mathcal{L}}(\mathbf{x}, \mathbf{y}, \xi) - \mathcal{L}(\mathbf{x}, \mathbf{y})) \frac{\xi^i}{c_h}. \tag{3}$$

Let $\tilde{\mathbf{e}}^i$ be the error signal computed by backpropagating the synthetic gradients:

$$\tilde{\mathbf{e}}^i = \begin{cases} \partial \mathcal{L}/\partial \hat{\mathbf{y}} \circ \sigma'(W^i \mathbf{h}^{i-1}), & i = N+1; \\ \left( (\hat{B}^{i+1})^T \tilde{\mathbf{e}}^{i+1} \right) \circ \sigma'(W^i \mathbf{h}^{i-1}), & 1 \leq i \leq N. \end{cases}$$

Then parameters $B^{i+1}$ are estimated by solving the least squares problem:

$$\hat{B}^{i+1} = \mathrm{argmin}_B \mathbb{E} \left\| B^T \tilde{\mathbf{e}}^{i+1} - \hat{\lambda}^i \right\|_2^2. \tag{4}$$

Under what conditions can we show that $\hat{B}^{i+1} \to W^{i+1}$ (with enough data)?

One way to find an answer is to define the synthetic gradient in terms of the system without noise added. Then $B^T \tilde{\mathbf{e}}$ is deterministic with respect to $\mathbf{x}, \mathbf{y}$ and, assuming $\tilde{\mathcal{L}}$ has a convergent power series around $\xi = 0$, we can write

$$\mathbb{E}(\hat{\lambda}^i | \mathbf{x}, \mathbf{y}) = \mathbb{E} \left( \frac{1}{c_h^2} \left[ \frac{\partial \mathcal{L}}{\partial h^i} (c_h \xi_j^i)^2 + \sum_{m=2}^{\infty} \frac{\mathcal{L}_{ij}^{(m)}}{m!} (c_h \xi_j^i)^{m+1} \right] | \mathbf{x}, \mathbf{y} \right)$$

$$= (W^{i+1})^T \mathbf{e}^{i+1} + \mathbb{E} \left( \frac{1}{c_h^2} \sum_{m=2}^{\infty} \frac{\mathcal{L}_{ij}^{(m)}}{m!} (c_h \xi_j^i)^{m+1} | \mathbf{x}, \mathbf{y} \right).$$

Taken together these suggest we can prove $\hat{B}^{i+1} \to W^{i+1}$ in the same way we prove consistency of the linear least squares estimator.

For this to work we must show the expectation of the Taylor series approximation (2) is well behaved. That is, we must show the expected remainder term of the expansion:

$$\mathcal{E}_j^i(c_h) = \mathbb{E} \left[ \frac{1}{c_h^2} \sum_{m=2}^{\infty} \frac{\mathcal{L}_{ij}^{(m)}}{m!} (c_h \xi_j^i)^{m+1} | \mathbf{x}, \mathbf{y} \right],$$

is finite and goes to zero as $c_h \to 0$. This requires some additional assumptions on the problem.

We make the following assumptions:

- A1: the noise $\xi$ is subgaussian,
- A2: the loss function $\mathcal{L}(\mathbf{x}, \mathbf{y})$ is analytic on $\mathcal{D}$,
- A3: the error matrices $\tilde{\mathbf{e}}^n (\tilde{\mathbf{e}}^n)^T$ are full rank, for $1 \leq n \leq N+1$,
- A4: the mean of the remainder and error terms is bounded:
$$\mathbb{E} \left[ \mathcal{E}^n(c_h) (\tilde{\mathbf{e}}^{n+1})^T \right] < \infty,$$
  for $1 \leq n \leq N$.

Consider first convergence of the final layer feedback matrix, $B^{N+1}$. In the final layer it is true that $\mathbf{e}^{N+1} = \tilde{\mathbf{e}}^{N+1}$.

**Theorem 1.** *Assume A1-4. For $\mathbf{g}_{FA}(\mathbf{h}^i, \tilde{\mathbf{e}}^{i+1}; B^{i+1}) = B^{i+1} \tilde{\mathbf{e}}^{i+1}$, then the least squares estimator*

$$(\hat{B}^{N+1})^T := \hat{\lambda}^N (\mathbf{e}^{N+1})^T \left( \mathbf{e}^{N+1} (\mathbf{e}^{N+1})^T \right)^{-1}, \tag{5}$$

*solves (4) and converges to the true feedback matrix, in the sense that:*

$$\lim_{c_h \to 0} \mathrm{plim}_{T \to \infty} \hat{B}^{N+1} = W^{N+1}.$$

*Proof.* Let $\mathcal{L}_{ij}^{(m)} := \frac{\partial^m \mathcal{L}}{\partial h_j^{im}}$. We first show that, under A1-2, the conditional expectation of the estimator (5) converges to the gradient $\mathcal{L}_{Nj}^{(1)}$ as $c_h \to 0$. For each $\hat{\lambda}_j^N$, by A2, we have the following series expanded around $\xi = 0$:

$$\hat{\lambda}_j^N = \frac{1}{c_h^2} \sum_{m=1}^{\infty} \frac{\mathcal{L}_{ij}^{(m)}}{m!} (c_h \xi_j^N)^{m+1}.$$

Taking a conditional expectation gives:

$$\mathbb{E}(\hat{\lambda}_j^N | \mathbf{x}, \mathbf{y}) = (W^{N+1})^T \mathbf{e}^{N+1} + \mathbb{E}\left[ \frac{1}{c_h^2} \sum_{m=2}^{\infty} \frac{\mathcal{L}_{Nj}^{(m)}}{m!} (c_h \xi_j^N)^{m+1} | \mathbf{x}, \mathbf{y} \right].$$

We must show the remainder term

$$\mathcal{E}^N(c_h) = \mathbb{E}\left[ \frac{1}{c_h^2} \sum_{m=2}^{\infty} \frac{\mathcal{L}_{Nj}^{(m)}}{m!} (c_h \xi_j^N)^{m+1} | \mathbf{x}, \mathbf{y} \right],$$

goes to zero as $c_h \to 0$. This is true provided each moment $\mathbb{E}((\xi_j^N)^m | \mathbf{x}, \mathbf{y})$ is sufficiently well-behaved. Using Jensen's inequality and the triangle inequality in the first line, we have that

$$\left| \mathcal{E}^N(c_h) \right| \leq \mathbb{E}\left[ \frac{1}{c_h^2} \sum_{m=2}^{\infty} \left| \frac{\mathcal{L}_{Nj}^{(m)}}{m!} \right| |c_h \xi_j^N|^{m+1} | \mathbf{x}, \mathbf{y} \right], \quad \forall (\mathbf{x}, \mathbf{y}) \in \mathcal{D}$$

$$[\text{monotone convergence}] \quad = \sum_{m=2}^{\infty} \left| \frac{\mathcal{L}_{Nj}^{(m)}}{m!} \right| (c_h)^{m-1} \mathbb{E}\left[ |\xi_j^N|^{m+1} \right]$$

$$[\text{subgaussian}] \quad \leq K \sum_{m=2}^{\infty} \left| \frac{\mathcal{L}_{Nj}^{(m)}}{m!} \right| (c_h)^{m-1} (\sqrt{m+1})^{m+1}$$

$$= \mathcal{O}(c_h) \quad \text{as } c_h \to 0. \tag{6}$$

With this in place, we have that the problem (4) is close to a linear least squares problem, since

$$\hat{\lambda}^N = (W^{N+1})^T \mathbf{e}^{N+1} + \mathcal{E}^N(c_h) + \eta^N, \tag{7}$$

with residual $\eta^N = \hat{\lambda}^N - \mathbb{E}(\hat{\lambda}^N | \mathbf{x}, \mathbf{y})$. The residual satisfies

$$\mathbb{E}\left( \mathbf{e}^{N+1} (\eta^N)^T \right) = \mathbb{E}(\mathbf{e}^{N+1} (\hat{\lambda}^N)^T - \mathbf{e}^{N+1} \mathbb{E}((\hat{\lambda}^N)^T | \mathbf{x}, \mathbf{y}))$$

$$= \mathbb{E}\left( \mathbf{e}^{N+1} (\hat{\lambda}^N)^T - \mathbb{E}\left( \mathbf{e}^{N+1} (\hat{\lambda}^N)^T | \mathbf{x}, \mathbf{y} \right) \right)$$

$$= 0. \tag{8}$$

This follows since $\mathbf{e}^{N+1}$ is defined in relation to the baseline loss, not the stochastic loss, meaning it is measurable with respect to $(\mathbf{x}, \mathbf{y})$ and can be moved into the conditional expectation.

From (7) and A3, we have that the least squares estimator (5) satisfies

$$(\hat{B}^{N+1})^T = (W^{N+1})^T + (\mathcal{E}^N(c_h) + \eta^N)(\mathbf{e}^{N+1})^T (\mathbf{e}^{N+1} (\mathbf{e}^{N+1})^T)^{-1}.$$

Thus, using the continuous mapping theorem

$$\underset{T \to \infty}{\text{plim}} (\hat{B}^{N+1})^T = (W^{N+1})^T + \left[ \underset{T \to \infty}{\text{plim}} \frac{1}{T} (\mathcal{E}^N(c_h) + \eta^N)(\mathbf{e}^{N+1})^T \right] \left[ \underset{T \to \infty}{\text{plim}} \frac{1}{T} \mathbf{e}^{N+1} (\mathbf{e}^{N+1})^T \right]^{-1}$$

$$[\text{WLLN}] \quad = (W^{N+1})^T + \mathbb{E}\left[ (\mathcal{E}(c_h) + \eta^N)(\mathbf{e}^{N+1})^T \right] \left[ \mathbb{E}(\mathbf{e}^{N+1} (\mathbf{e}^{N+1})^T) \right]^{-1}$$

$$[\text{Eq. (8)}] \quad = (W^{N+1})^T + \mathbb{E}\left[ \mathcal{E}(c_h)(\mathbf{e}^{N+1})^T \right] \left[ \mathbb{E}(\mathbf{e}^{N+1} (\mathbf{e}^{N+1})^T) \right]^{-1}$$

$$[\text{A4 and Eq. (6)}] \quad = (W^{N+1})^T + \mathcal{O}(c_h).$$

Then we have:

$$\lim_{c_h \to 0} \underset{T \to \infty}{\text{plim}} \hat{B}^{N+1} = W^{N+1}.$$

$\square$

We can use Theorem 1 to establish convergence over the rest of the layers of the network when the activation function is the identity.

**Theorem 2.** *Assume A1-4. For* $\mathbf{g}_{FA}(\mathbf{h}^i, \tilde{\mathbf{e}}^{i+1}; B^{i+1}) = B^{i+1}\tilde{\mathbf{e}}^{i+1}$ *and* $\sigma(x) = x$, *the least squares estimator*

$$(\hat{B}^n)^T := \hat{\lambda}^{n-1}(\tilde{\mathbf{e}}^n)^T \left(\tilde{\mathbf{e}}^n(\tilde{\mathbf{e}}^n)^T\right)^{-1} \qquad 1 \leq n \leq N+1, \tag{9}$$

*solves (4) and converges to the true feedback matrix, in the sense that:*

$$\lim_{c_h \to 0} \operatorname*{plim}_{T \to \infty} \hat{B}^n = W^n, \qquad 1 \leq n \leq N+1.$$

*Proof.* Define

$$\tilde{W}^n(c) := \operatorname*{plim}_{T \to \infty} \hat{B}^n,$$

assuming this limit exists. From Theorem 1 the top layer estimate $\hat{B}^{N+1}$ converges in probability to $\tilde{W}^{N+1}(c)$.

We can then use induction to establish that $\hat{B}^j$ in the remaining layers also converges in probability to $\tilde{W}^j(c)$. That is, assume that $\hat{B}^j$ converge in probability to $\tilde{W}^j(c)$ in higher layers $N + 1 \geq j > n$. Then we must establish that $\hat{B}^n$ also converges in probability.

To proceed it is useful to also define

$$\tilde{\tilde{\mathbf{e}}}(c)^n := \begin{cases} \partial\mathcal{L}/\partial\hat{\mathbf{y}} \circ \sigma'(W^i\mathbf{h}^{i-1}), & i = N+1; \\ \left((\tilde{W}^{i+1}(c))^T\tilde{\tilde{\mathbf{e}}}^{i+1}\right) \circ \sigma'(W^i\mathbf{h}^{i-1}), & 1 \leq i \leq N, \end{cases}$$

as the error signal backpropagated through the converged (but biased) weight matrices $\tilde{W}(c)$. Again it is true that $\tilde{\tilde{\mathbf{e}}}^{N+1} = \mathbf{e}^{N+1}$.

As in Theorem 1, the least squares estimator has the form:

$$(\hat{B}^n)^T = \hat{\lambda}^{n-1}(\tilde{\mathbf{e}}^n)^T \left(\tilde{\mathbf{e}}^n(\tilde{\mathbf{e}}^n)^T\right)^{-1}.$$

Thus, again by the continuous mapping theorem:

$$\operatorname*{plim}_{T \to \infty} (\hat{B}^n)^T = \left[\operatorname*{plim}_{T \to \infty} \frac{1}{T}\hat{\lambda}^{n-1}(\tilde{\mathbf{e}}^n)^T\right] \left[\operatorname*{plim}_{T \to \infty} \frac{1}{T}\tilde{\mathbf{e}}^n(\tilde{\mathbf{e}}^n)^T\right]^{-1}$$

$$= \left[\operatorname*{plim}_{T \to \infty} \frac{1}{T}\hat{\lambda}^{n-1}(\mathbf{e}^{N+1})^T \hat{B}^{N+1} \cdots \hat{B}^{n+1}\right] \left[\operatorname*{plim}_{T \to \infty} \frac{1}{T}\tilde{\mathbf{e}}^n(\tilde{\mathbf{e}}^n)^T\right]^{-1}$$

In this case continuity again allows us to separate convergence of each term in the product:

$$\operatorname*{plim}_{T \to \infty} \frac{1}{T}\hat{\lambda}^{n-1}(\mathbf{e}^{N+1})^T \hat{B}^{N+1} \cdots \hat{B}^{n+1} = \left[\operatorname*{plim}_{T \to \infty} \frac{1}{T}\hat{\lambda}^{n-1}(\mathbf{e}^{N+1})^T\right] \left[\operatorname*{plim}_{T \to \infty} \hat{B}^{N+1}\right] \cdots \left[\operatorname*{plim}_{T \to \infty} \hat{B}^{n+1}\right] \tag{10}$$

$$= \mathbb{E}(\hat{\lambda}^{n-1}(\mathbf{e}^{N+1})^T)W^{N+1}(c) \cdots W^{n+1}(c),$$

$$= \mathbb{E}(\hat{\lambda}^{n-1}(\tilde{\tilde{\mathbf{e}}}^n(c))^T)$$

using the weak law of large numbers in the first term, and the induction assumption for the remaining terms. In the same way

$$\operatorname*{plim}_{T \to \infty} \frac{1}{T}\tilde{\mathbf{e}}^n(\tilde{\mathbf{e}}^n)^T = \mathbb{E}(\tilde{\tilde{\mathbf{e}}}^n(c)(\tilde{\tilde{\mathbf{e}}}^n(c))^T).$$

Note that the induction assumption also implies $\lim_{c \to 0} \tilde{\tilde{\mathbf{e}}}^n(c) = \mathbf{e}^n$. Thus, putting it together, by A3, A4 and the same reasoning as in Theorem 1 we have the result:

$$\lim_{c_h \to 0} \operatorname*{plim}_{T \to \infty} (\hat{B}^n)^T = \lim_{c \to 0} \left[(W^n)^T\mathbb{E}(\mathbf{e}^n(\tilde{\tilde{\mathbf{e}}}^n(c))^T) + \mathbb{E}(\mathcal{E}^{n-1}(c)(\tilde{\tilde{\mathbf{e}}}^n(c))^T)\right] \left[\mathbb{E}(\tilde{\tilde{\mathbf{e}}}^n(c)(\tilde{\tilde{\mathbf{e}}}^n(c))^T)\right]^{-1}$$

$$= (W^n)^T.$$

$\square$

**Corollary 1.** *Assume A1-4. For* $\mathbf{g}_{DFA}(\mathbf{h}^i, \tilde{\mathbf{e}}^{N+1}; B^{i+1}) = B^{i+1}\tilde{\mathbf{e}}^{N+1}$ *and* $\sigma(x) = x$, *the least squares estimator*

$$(\hat{B}^n)^T := \hat{\lambda}^{n-1}(\tilde{\mathbf{e}}^{N+1})^T \left(\tilde{\mathbf{e}}^{N+1}(\tilde{\mathbf{e}}^{N+1})^T\right)^{-1} \qquad 1 \leq n \leq N+1, \tag{11}$$

*solves (4) and converges to the true feedback matrix, in the sense that:*

$$\lim_{c_h \to 0} \operatorname*{plim}_{T \to \infty} \hat{B}^n = \prod_{j=N+1}^{n} W^j, \qquad 1 \leq n \leq N+1.$$

*Proof.* For a deep linear network notice that the node perturbation estimator can be expressed as:

$$\hat{\lambda}^i = (W^{n+1} \cdots W^{N+1})^T \mathbf{e}^{N+1} + \mathcal{E}^n(c_h) + \eta^n, \tag{12}$$

where the first term represents the true gradient, given by the simple linear backpropagation, the second and third terms are the remainder and a noise term, as in Theorem 1. Define

$$V^n := \prod_{j=N+1}^{n} W_j.$$

Then following the same reasoning as the proof of Theorem 1, we have:

$$\operatorname*{plim}_{T\to\infty} (\hat{B}^{n+1})^T = (V^{n+1})^T + \left[ \operatorname*{plim}_{T\to\infty} \frac{1}{T}(\mathcal{E}^n(c_h) + \eta^n)(\mathbf{e}^{N+1})^T \right] \left[ \operatorname*{plim}_{T\to\infty} \frac{1}{T} \mathbf{e}^{N+1}(\mathbf{e}^{N+1})^T \right]^{-1}$$

$$= (V^{n+1})^T + \mathbb{E}\left[ (\mathcal{E}(c_h) + \eta^n)(\mathbf{e}^{N+1})^T \right] \left[ \mathbb{E}(\mathbf{e}^{N+1}(\mathbf{e}^{N+1})^T) \right]^{-1}$$

$$= (V^{n+1})^T + \mathbb{E}\left[ \mathcal{E}(c_h)(\mathbf{e}^{N+1})^T \right] \left[ \mathbb{E}(\mathbf{e}^{N+1}(\mathbf{e}^{N+1})^T) \right]^{-1}$$

$$= (V^{n+1})^T + \mathcal{O}(c_h).$$

Then we have:

$$\lim_{c_h \to 0} \operatorname*{plim}_{T\to\infty} \hat{B}^{n+1} = V^{n+1}.$$

$\square$

## Discussion of assumptions

It is worth making the following points on each of the assumptions:

- A1. In the paper we assume $\xi$ is Gaussian. Here we prove the more general result of convergence for any subgaussian random variable.

- A2. In practice this may be a fairly restrictive assumption, since it precludes using relu non-linearities. Other common choices, such as hyperbolic tangent and sigmoid non-linearities with an analytic cost function do satisfy this assumption, however.

- A3. It is hard to establish general conditions under which $\tilde{\mathbf{e}}^n(\tilde{\mathbf{e}}^n)^T$ will be full rank. While it may be a reasonable assumption in some cases.

Extensions of Theorem 2 to a non-linear network may be possible. However, the method of proof used here is not immediately applicable because the continuous mapping theorem can not be applied in such a straightforward fashion as in Equation (10). In the non-linear case the resulting sums over all observations are neither independent or identically distributed, which makes applying any law of large numbers complicated.

