# OpenReview forum: "Learning to solve the credit assignment problem"
_NeurIPS.cc/2019/Workshop/Neuro_AI — Real Neurons & Hidden Units @ NeurIPS 2019 Poster_

### Official Review · AnonReviewer2 · 2019-09-23
**A step towards credit assignment without weight transport**

**Clarity:** 5

**Comment:**

The similarity between this work and Akrout et al. (2019) is definitely large. Would be curious to hear the author’s thoughts on the potential advantages / disadvantages of their method in comparison.

**Category:**

Common question to both AI & Neuro

**Clarity Comment:**

The method and benchmarks being performed are described clearly and with reference to the relevant literature.

**Evaluation:**

4: Very good

**Importance:**

4: Very important

**Importance Comment:**

The current work presents an algorithm for neural network training using node perturbation that does not rely on weight transport and performs well on a number of difficult machine learning problems. These methods are essential for neuroscience and AI and will hopefully make solid testable predictions in the near future.

**Intersection:**

4: High

**Intersection Comment:**

How real and artificial neural networks can learn without direct access to synaptic weight information from other neurons (“weight transport”) is an essential question in both neuroscience and AI.

**Rigor Comment:**

The results of node perturbation for MNIST, auto-encoding MNIST, and CIFAR are convincing. Authors show the average of multiple runs and over different noise levels. Where the method has drawbacks (noise requirements, baseline loss, separate feedforward and feedback learning), the authors have clearly pointed to ways these requirements are in line with biology, or could be removed in future work.

**Technical Rigor:**

4: Very convincing

---

### Official Review · AnonReviewer3 · 2019-09-27

**Clarity:** 3

**Comment:**

I don't understand the need for the noise perturbations. This work proposes updating the backwards weights with (B^T e - lambda) e^T, and states that doing so will cause them to converge towards the transpose of the forward weights. Wouldn't it be simpler, and require a less complex circuit, simply to update the backwards weights with h^{i-1} e^T? (as is proposed in [Kolen and Pollack, 1994]). In this case, foward and reverse weights will also converge towards each other. It seems like doing this by injecting noise instead of just using the forward activations requires both a more complex, and noisier, circuit.

Also, if every unit is simultaneously injecting noise, it's not obvious to me that this will scale better with number of units than RL -- I suspect the scaling will be exactly the same, since noise contributions from different units will interfere with each other.

(should cite evolutionary strategies for your functional form for lambda)


**Category:**

AI->Neuro

**Clarity Comment:**

This was clearly written, but seemed unnecessarily complex.


**Evaluation:**

3: Good

**Importance:**

2: Marginally important

**Importance Comment:**

Understanding how learning occurs in the brain is extremely important. Understanding how the brain could implement backprop is also important. This approach seems technically correct, but inefficient with potential scaling issues -- it seems unlikely it will change how readers think about learning in the brain.

**Intersection:**

4: High

**Intersection Comment:**

This work focuses on porting the idea of backprop from AI to neuro.


**Rigor Comment:**

I believe all claims are correct.


**Technical Rigor:**

4: Very convincing

---

### Official Review · AnonReviewer1 · 2019-09-27
**Rich experimental analysis supports an improvement over relying on feedback alignment**

**Clarity:** 2

**Category:**

Common question to both AI & Neuro

**Clarity Comment:**

I understood the methods section up until "we will use the noisy response to estimate gradients", which is why I don't have a good sense of how this approach will scale (see reviewer 3's comment about simultaneous noise injection). Other than this, the paper is interesting, well written, and well organized.

**Evaluation:**

4: Very good

**Importance:**

4: Very important

**Importance Comment:**

Learning effective backward-pass weights is an important step towards biologically plausible learning of difficult tasks in ML.

**Intersection:**

5: Outstanding

**Intersection Comment:**

Learning without weight transport is of interest to members of both communities.

**Rigor Comment:**

The experiments and visualizations are rich and convincing. A learning-based approach to credit assignment seems to be clearly better than relying on feedback alignment. I agree with reviewer 3 that a discussion of training signal variance scaling would be helpful, and I agree with reviewer 2 that comparisons to more related approaches would be interesting.

**Technical Rigor:**

3: Convincing

---

### Decision · Program_Chairs · 2019-10-02

Accept (Poster)